# Custom Gene Panel Analysis Identifies Novel Polymorphisms Associated with Clopidogrel Response in Patients Undergoing Percutaneous Coronary Intervention with Stent

**DOI:** 10.3390/ijms26199766

**Published:** 2025-10-07

**Authors:** Alba Antúnez-Rodríguez, Sonia García-Rodríguez, Ana Pozo-Agundo, Jesús Gabriel Sánchez-Ramos, Eduardo Moreno-Escobar, José Matías Triviño-Juárez, María Jesús Álvarez-Cubero, Luis Javier Martínez-González, Cristina Lucía Dávila-Fajardo

**Affiliations:** 1Center for Genomics and Oncological Research (GENYO), Pfizer—University of Granada—Junta de Andalucía, Avenida de la Ilustración 114, 18016 Granada, Spain; e.albant@go.ugr.es (A.A.-R.); garciarodriguez.sonia@gmail.com (S.G.-R.); ana222pozo2@hotmail.com (A.P.-A.); mjesusac@ugr.es (M.J.Á.-C.); luisjavier.martinez@genyo.es (L.J.M.-G.); 2Instituto de Investigación Biosanitaria (ibs.GRANADA), Avenida de Madrid 15, Pabellón de Consultas Externas, 2ª Planta (Antigua Área de Dirección), 18012 Granada, Spain; 3Cardiology Department, Hospital Universitario Clínico San Cecilio, Avenida de la Innovación s/n, 18016 Granada, Spain; jgsr525@gmail.com (J.G.S.-R.); eduroc6868@gmail.com (E.M.-E.); 4Department of Radiology and Physical Medicine, Faculty of Medicine, University of Granada, Avenida Doctor Jesús Candel Fábregas 11, 18016 Granada, Spain; jmtjuarez@ugr.es; 5Department of Biochemistry and Molecular Biology III and Inmunology, Faculty of Medicine, University of Granada, Avenida Doctor Jesús Candel Fábregas 11, 18016 Granada, Spain; 6Pharmacy Department, Hospital Universitario Virgen de las Nieves, Avenida de las Fuerzas Armadas 2, 18014 Granada, Spain

**Keywords:** dual antiplatelet therapy, cardiovascular drugs, acute coronary syndrome, clopidogrel response, atherosclerosis, pharmacogenomics, personalized medicine

## Abstract

Clopidogrel is widely used as an antiplatelet therapy for acute coronary syndrome (ACS) patients undergoing percutaneous coronary intervention (PCI). Genetic factors influence variability in clopidogrel response, with non-functional *CYP2C19* alleles increasing the risk of major adverse cardiovascular events (MACEs). While *CYP2C19* genotype-guided therapy after PCI improves outcomes, MACEs persist at variable rates. Pharmacogenomics (PGx) has primarily focused on genes related to drug metabolism, but therapeutic failure may stem from individual disease predisposition. This study aims to identify novel genetic variants underlying adverse events after PCI despite PGx-guided therapy. A custom sequencing panel was analyzed in 244 ACS-PCI-stent patients and 99 controls without cardiovascular (CV) disease. Association analysis was performed independent of treatment and by prescribed treatment (clopidogrel or prasugrel), complemented by random forest models to predict risk during antiplatelet therapy. No polymorphism reached genomic significance, but in clopidogrel-treated patients, rs2472434 in *ABCA1*, related to altered lipid metabolism, was strongly associated with secondary CV events (*p* = 1.7 × 10^−3^). Variants in the clopidogrel pathway, including *CYP2C19*, *ABCB1*, and *UGT2B7*, were also identified and may influence clopidogrel response. Predictive models incorporating these variants effectively discriminated patients with and without events (*p* = 0.02445). Our findings support combined genotyping of *CYP2C19* loss-of-function and *ABCB1 C3435T* variants to guide antiplatelet therapy and suggest additional targets, such as rs2472434 (*ABCA1*) and rs7439366 (*UGT2B7*), to improve risk prediction of adverse CV events. Therefore, the unexplained variability in clopidogrel response may be due to disease pathogenesis itself, highlighting the need for a paradigm shift in PGx studies.

## 1. Introduction

Dual antiplatelet therapy (DAPT), combining acetylsalicylic acid and a platelet adenosine diphosphate receptor (P2Y_12_) inhibitor, is the standard of care for patients undergoing percutaneous coronary intervention (PCI), particularly those with acute coronary syndrome (ACS), to prevent subsequent postprocedural thrombotic events. Among P2Y_12_ inhibitors, clopidogrel remains one of the most widely used antiplatelet agents in the treatment of ACS patients due to its efficacy and lower bleeding risk compared to newer alternatives such as prasugrel and ticagrelor [1].

Clopidogrel is a prodrug that, once absorbed in the duodenum—where its absorption and bioavailability are limited by P-glycoprotein, an intestinal efflux pump encoded by the *ABCB1* gene—requires hepatic biotransformation by cytochrome P450 (CYP) enzymes, particularly CYP2C19, to produce its active metabolite, which inhibits ADP-induced platelet activation and aggregation. Despite its efficacy, there is significant inter-individual variability in clopidogrel response, with 20–40% of patients experiencing high on-treatment platelet reactivity and an increased risk of major adverse cardiac events (MACEs) following PCI [2,3].

The variability in clopidogrel efficacy has been associated with non-genetic factors such as age, sex, body mass index (BMI), comorbidities, or drug–drug interactions, among others. However, clopidogrel response is highly heritable and has also been associated with genetic polymorphisms affecting both drug pharmacokinetics and pharmacodynamics. A genome-wide association study conducted in a large Amish population revealed that ~70% of the variability in clopidogrel response may be due to genetic factors, with the *CYP2C19***2* allele (rs4244285) being the major genetic determinant. Carriers of this loss-of-function (LoF) allele are less efficient at converting clopidogrel to its active metabolite, resulting in reduced platelet inhibition and an increased risk of ischemic events, particularly after PCI. However, this variant accounts for only 12% of the overall variation in platelet reactivity, suggesting the presence of other possible genetic contributions [4,5].

The association between *CYP2C19* LoF alleles and secondary adverse events in clopidogrel-treated patients is well established, and accumulating evidence from both observational and randomized studies has demonstrated the clinical benefit of using the *CYP2C19* genotype to guide the selection of prasugrel or ticagrelor in *CYP2C19* LoF allele carriers (mainly **2* and **3*) after PCI [6,7,8]. However, despite the clinical relevance of *CYP2C19* genotyping when prescribing this type of drug, MACEs continue to occur at variable rates (10% in our population; consult Appendix A, p. 5 for more information) [9]. This suggests that prescribing based solely on known *CYP2C19* LoF alleles may not be sufficient to optimize clinical outcomes. For this reason, numerous studies have attempted to identify additional genetic determinants of clopidogrel response, often through candidate gene approaches (e.g., *ABCB1*, *CES1* or *PON1*) [3,10]. However, discrepant and inconclusive results regarding the effect of other polymorphisms beyond those in *CYP2C19* on clopidogrel response have limited use in clinical practice [4,11].

Pharmacogenomics (PGx) focuses on exploring the role of genetic variation in drug response. To date, most PGx studies have focused primarily on the analysis of pharmacokinetic (absorption, distribution, metabolism, excretion) and pharmacodynamic (receptors or molecular targets) mechanisms underlying inter-individual variability in drug response [12,13]. However, therapeutic failure may be due to an individual’s predisposition to the disease itself, not just suboptimal metabolism of the drugs used to treat the disease. Therefore, the aim of this study was to evaluate a customized gene sequencing panel, including both PGx and cardiovascular (CV) disease-related genes, to identify novel undescribed genetic variants associated with the development of recurrent adverse events during one-year follow-up, despite PGx-guided antiplatelet therapy.

## 2. Results

### 2.1. Patient Characteristics

A total of 343 patients from Granada (Spain) were included in this study, 244 with ACS undergoing PCI and 99 controls without structural CV disease (Table 1). The mean age of the cohort was 65.9 ± 11.1 years, and the mean BMI was 28.6 ± 4.6. Among ACS patients, the presence of previous CV history was associated with the incidence of subsequent CV events (*p* = 0.008).

After hospitalization, all ACS patients were prescribed DAPT, which included acetylsalicylic acid and an antiplatelet agent (guided or not by genetic testing). As shown in Table 2, clopidogrel was prescribed to 169 patients (69.3%), prasugrel to 73 (29.9%), and ticagrelor to 2 (0.8%) (ticagrelor was approved during the course of the previous study conducted by our group). When comparing between groups, ACS patients who experienced a secondary CV event (G1) were more likely to be prescribed clopidogrel (75.2%) and less likely to be prescribed prasugrel (22.9%) than patients who did not experience an event (G2) (64.4% and 35.6%, respectively). This is explained by the fact that all patients in the G2 group were in the PGx intervention arm, where the antiplatelet agent was prescribed based on genotype results. In contrast, the G1 group included patients in both the intervention and non-intervention arms, which explains the observed difference (*p* = 0.035). In addition, 13.8% of G1 patients received DAPT for less than 12 months due to clinical decisions (e.g., risk/benefit considerations or the need for triple antiplatelet therapy), whereas 97.8% of G2 patients completed a full year of DAPT. For a more detailed description of clinical and demographic characteristics and other concomitant treatments, see Antúnez-Rodríguez et al. [2].

### 2.2. Two-Stage Association Study

In the first analysis of association, we established three comparisons:○“Intervention vs. non-intervention” (see Appendix A, p. 7)○“Event vs. non-event” (see Appendix A, p. 9 or in more detail in the publication by Antúnez-Rodríguez et al. [2])○“Case vs. control” (see Appendix A, p. 11)

The results of this first phase suggest that in ACS-PCI-stent patients, variants in key genes involved in lipid metabolism, vascular inflammation, and atherosclerotic plaque stability may play an important role in predisposing to secondary CV events (Table 3). However, the exact mechanisms by which these variants influence the risk of developing ischemic and/or hemorrhagic events are not fully understood, underscoring the need for further studies to better elucidate their impact and clinical relevance.

In a second association analysis, we again evaluated the phenotype related to the development of secondary CV events (“event vs. non-event” comparison), but in this case we decided to stratify according to the antiplatelet agent prescribed. Our objective was to test whether there are differential variants related to the metabolism of these drugs, which could be used to personalize treatment and further reduce the incidence of events in ACS-PCI-stent patients.

#### 2.2.1. Secondary CV Events After Clopidogrel Treatment

Among ACS-PCI-stent patients taking the antiplatelet drug clopidogrel, we compared those who experienced a secondary CV event after treatment with similarly treated patients who did not. Our aim was to determine whether there were any PGx variants that could be considered to further reduce the incidence of MACEs and/or hemorrhagic events.

After performing the association analysis for the phenotype related to the occurrence of secondary CV events in clopidogrel-treated patients, no locus reached the genome-wide significance threshold (lowest *p*-value = 4.6 × 10^−4^) in the model adjusted for age, gender, and principal components (Figure 1).

On chromosome 10, two polymorphisms in the *CYP2C19* gene (rs4244285 and rs4986894, in complete linkage disequilibrium (LD)) were associated with an increased risk of MACEs and/or hemorrhagic events (*p* = 4.6 × 10^−4^) (Table 4). However, as the group of patients without an event (G2) consisted only of individuals from the intervention arm (genotyped for polymorphisms in *CYP2C19* and *ABCB1*), this result was to be expected, since no patient with the *A*-risk allele for rs4244285 (*CYP2C19***2*, c.681G>A) was prescribed clopidogrel.

On the other hand, the rs2472434 variant in *ABCA1* (A>C), which proved to be the most significant polymorphism in the “event vs. non-event” comparison regardless of the treatment received [2], was also among the most significant single nucleotide polymorphisms (SNPs) when considering the prescribed drug. In this case, patients taking clopidogrel who were also carriers of the *C* allele had a higher incidence of secondary CV events during 1-year follow-up compared to non-carriers (0.34 vs. 0.19; β = −0.88; *p* = 1.7 × 10^−3^). Similarly, the ancestral *G* allele of the rs17618244 locus at *KLB* (G>A) was identified as suggestively associated with the incidence of events after clopidogrel treatment (β = 0.88; *p* = 1.9 × 10^−3^).

Despite these results, we were really interested in determining whether there were genetic variants within the genes described as being involved in clopidogrel metabolism that affected the pharmacokinetics and pharmacodynamics of the drug. Therefore, the results of the association analysis were limited to the genes described by PharmGKB (Appendix A).

On chromosome 7, a polymorphism in the *ABCB1* gene (rs2235048, c.3699+80C>T), in complete LD with *ABCB1* c.3435C>T, was associated with a lower incidence of secondary CV events in patients prescribed clopidogrel (β = 0.63; *p* = 0.03322) (Table 5). As in the case of *CYP2C19***2*, since the group of patients without an event (G2) consisted entirely of individuals from the PGx intervention arm, this result was to be expected, as no individuals with the *TT* risk genotype for rs1045642 (c.3435 C>T) was prescribed clopidogrel. However, two polymorphisms in the *UGT2B7* gene (rs28365062 and rs7439366, unlinked) and four in the *P2RY12* gene (rs1907637, rs6809699, rs2046934, and rs10935838, these three in complete LD with each other) were similarly associated with a lower incidence of MACEs and/or hemorrhagic events in patients treated with clopidogrel.

Taken together, the results of this analysis revealed several genetic variants of interest that may influence the response to clopidogrel therapy in ACS-PCI-stent patients. Consequently, we decided to include these variants in subsequent analyses (see Section 2.3) to evaluate their potential as possible biomarkers capable of optimizing treatment personalization and improving therapeutic efficacy in these patients.

#### 2.2.2. Secondary CV Events After Prasugrel Treatment

As in the previous section, among ACS-PCI-stent patients taking the antiplatelet agent prasugrel, we compared those who experienced a post-treatment secondary CV event with those who did not during the 1-year follow-up period. Again, our aim was to determine whether there were any PGx variants that could be considered to further reduce the incidence of MACEs and/or hemorrhagic events.

Once the association analysis was performed for the phenotype related to the occurrence of secondary CV events in prasugrel-treated patients, no locus reached the genome-wide significance threshold (lowest *p*-value = 2.6 × 10^−4^) in the model adjusted for age, gender, and principal components (Figure 2).

On chromosome 21, the rs13047599 polymorphism in the *SON* gene (c.4723C>T, p.Arg1575Cys) was associated with the occurrence of MACEs and/or hemorrhagic events (Table 6). In this case, carriers of the alternative *T* allele who took prasugrel had a higher incidence of secondary CV events compared with non-carriers (0.84 vs. 0.57; β = −1.31; *p* = 5.7 × 10^−4^). Similarly, on chromosome 3, the rs3732511 (c.1155G>C) locus in the *ARHGEF3* gene was also associated with a higher incidence of secondary CV events in carriers of the alternative *G* allele who were additionally treated with prasugrel (0.28 vs. 0.07; β = −1.81; *p* = 6.2 × 10^−4^).

Interestingly, regarding the most significant polymorphisms in the “event vs. non-event” comparison regardless of the treatment received, the results of this sub-analysis suggest that rs2472434 in *ABCA1* (c.543+711T>G) remains associated with a higher risk of secondary CV events in prasugrel-treated patients. In this case, carriers of the *C* allele had a higher incidence of events during 1-year follow-up compared to non-carriers (0.40 vs. 0.20; β = −1.27; *p* = 5.7 × 10^−3^). Similarly, the ancestral *C* allele of rs3827066 in *ZNF335* (c.2442+202G>A) was suggestively associated with the development of MACEs and/or hemorrhagic events in this setting (β = 1.05; *p* = 0.0288).

As with clopidogrel, we were really interested in determining whether there were genetic variants within the genes described as being involved in prasugrel metabolism that affected the pharmacokinetics and pharmacodynamics of the drug. The results of this analysis can be found in the Appendix A, p. 13.

### 2.3. Random Forest Models

Based on the results of the various association analyses and in combination with the most relevant clinical variables, a random forest analysis was performed to assess the predictive ability of each model and to determine the relative importance of each “predictor variable” in the susceptibility to develop secondary CV events during follow-up.

#### 2.3.1. Secondary CV Events Regardless of Prescribed Therapy

For the “event vs. non-event” comparison, regardless of the antiplatelet agent received, random forest analysis revealed that the rs2472434 (*ABCA1*) genotype was the main predictor variable from a ranked list of variables according to their importance in the classification scheme (Figure 3). When the full model was evaluated, the rs2472434 (*ABCA1*) genotype, along with other clinical variables such as BMI, age, and principal diagnosis at admission, retained its relevance in discriminating between patients with and without an event, strengthening its predictive value.

The prognostic performance of the variables included in the established model was then evaluated. The receiver operating characteristic (ROC) curve showed an area under the curve (AUC) value of 0.625, indicating a moderate discriminatory ability (Figure 4). The model was able to correctly classify 60% of the patients, with a sensitivity of 43.75% and a specificity of 75.76%. The obtained *p*-value (*p* = 0.077) was close to the threshold of statistical significance (*p* < 0.05), as the model tended to classify more accurately patients with no adverse events during the follow-up period (G2 group). However, in the case of patients with an event (G1 group), the proposed model did not exceed the expected accuracy by hazard.

#### 2.3.2. Secondary CV Events Following Clopidogrel Treatment

Random forest analysis for the “event vs. non-event” comparison in clopidogrel-treated patients identified the “clopidogrel resistance genotype”, defined by variants in the *CYP2C19* (**2* and **3*) and *ABCB1* (*C3435T*) genes, as the most important predictor variable within the model, as it clearly stood out in the classification scheme (Figure 5). In addition to the resistance genotype, other genetic variants such as those of the *UGT2B7* (rs7439366) and *ABCA1* (rs2472434) genes also showed a remarkable ability to discriminate between groups (multidimensional scaling plots obtained from the training set are shown in Appendix A). Likewise, when the full model was evaluated, these genetic variants, along with other clinical variables such as BMI and age, retained their relevance in discriminating between the study groups.

When assessing the prognostic performance of the proposed model, it showed an AUC value of 0.713, correctly classifying 65.22% of the patients, with a sensitivity of 45.83% and a specificity of 86.36% (Figure 6). The *p*-value of 0.024 indicates that the established model has a high predictive ability to discriminate between patients with and without an event, with a tendency to better classify those who did not experience a secondary CV event after clopidogrel treatment (G2 group).

#### 2.3.3. Secondary CV Events Following Prasugrel Treatment

No relevant findings were observed in this section. Detailed data supporting this observation are provided in Appendix A, p. 16.

## 3. Discussion

To our knowledge, the current study is the first to design a personalized gene panel that includes genes involved in the metabolism and transport of antiplatelet agents as well as genes associated with the development of CV disease. This panel aimed to identify novel genetic variants that will further our understanding of the genetic factors underlying the occurrence of MACEs and/or hemorrhagic events in patients receiving DAPT after PCI despite following a prescription guided by genetic testing.

Exploratory variant analysis identified rs2472434 (c.543 +711T>G), an intronic variant in the *ABCA1* gene, as “suggestively” associated with these adverse events during one-year follow-up, both independent of treatment (*p* = 1.0 × 10^−4^) and when considering the type of drug prescribed (*p* = 1.7 × 10^−3^ for clopidogrel and *p* = 5.7 × 10^−3^ for prasugrel). Random forest analysis further highlighted the rs2472434 genotype as the primary predictive variable in our model, reinforcing its potential role in the increased incidence of secondary CV events following DAPT. *ABCA1* encodes an integral membrane protein that is widely expressed in many tissues and cells, including enterocytes, hepatocytes, and macrophages. This transporter plays a key role in cholesterol homeostasis and prevention of coronary atherosclerosis by mediating the cellular efflux of cholesterol and phospholipids, thereby facilitating high-density lipoprotein (HDL) synthesis and promoting reverse cholesterol transport (RCT), which limits atherosclerotic plaque formation and progression [15]. Genome-wide association studies have identified several common SNPs in non-coding regions of *ABCA1* (e.g., in the promoter, intron 1, and 5′ untranslated region) that may affect the proper regulation of *ABCA1* expression and the severity of atherosclerosis [2,16]. Although the literature on rs2472434 in the context of ACS is limited, GTEx analysis showed that the *C* risk allele is significantly associated with lower hepatic expression of *ABCA1* (Appendix A). This reduction could increase the risk of atherosclerosis and recurrent events by reducing cholesterol efflux and consequently promoting lipid accumulation in atherosclerotic plaques. Consistent with this hypothesis, a transcriptional analysis by Suresh et al. [17] revealed that *ABCA1* was downregulated in the transcriptome of ACS patients with recurrent events, highlighting that alterations in cholesterol transport pathways during initial ACS presentation are associated with increased disease severity. In addition, HDL has been implicated in increased platelet reactivity after discontinuation of P2Y_12_ receptor antagonist therapy. In particular, the rebound phenomenon observed after clopidogrel discontinuation in patients receiving DAPT has been associated with an altered lipid profile. Loss of the P2Y_12_ receptor blockade may be an important prothrombotic and proinflammatory trigger, particularly in patients at increased CV risk, such as those with low HDL levels [2,18]. However, the relationships between HDL levels, platelet aggregation, and associated adverse events are complex and remain to be fully elucidated. Therefore, functional validation studies assessing the impact of this *ABCA1* variant on susceptibility to events during antiplatelet treatment are required prior to clinical implementation.

When patients were stratified based on their prescribed antiplatelet drug, carriers of the “resistance genotype”—defined by variants in the *CYP2C19* (*A* allele for rs4244285) and *ABCB1* (*TT* genotype for rs1045642) genes—who were treated with clopidogrel (non-intervention group) had a higher incidence of MACEs and/or hemorrhagic events during one-year follow-up. Random forest analysis further identified this “resistance genotype” as the most important predictor of adverse events after stent implantation. These findings are consistent with the PGx analysis of the TRITON-TIMI 38 trial, which showed that both *CYP2C19* LoF alleles and the *ABCB1 TT* genotype were independent predictors of MACEs in clopidogrel-treated ACS-PCI patients (pooled HR = 1.97; 95% CI: 1.38–2.82; *p* = 0.0002) [19]. Subsequent studies have confirmed *CYP2C19* as the primary genetic determinant of variable clopidogrel response, although the widely studied *CYP2C19***2* variant explains only 14% of response variability, suggesting other contributing factors [20]. The *ABCB1* gene has also been associated with variable response to clopidogrel, but discrepancies remain in the literature regarding the effect of the *ABCB1 C3435T* variant (rs1045642), largely due to studies evaluating the metabolizing enzyme and drug transporter in isolation rather than evaluating the combined effect of both genes [10]. Our findings are consistent with previous studies showing that the combination of *CYP2C19* and *ABCB1* polymorphisms may improve the prediction of adverse clinical outcomes in ACS patients undergoing PCI, rather than *CYP2C19* genotyping alone as suggested by guidelines [21]. For instance, Galeazzi et al. [22] found that carriers of the *A* allele for *CYP2C19***2* and the *TT* genotype for *ABCB1 C3435T* had an increased risk of thrombotic events (OR = 1.26; 95% CI: 1.099–1.45; *p* = 0.027), supporting the hypothesis that the combined effect of both polymorphisms is more predictive than analyzing either gene individually. Similarly, the RAPID STEMI study [23] showed that both *CYP2C19* LoF allele and *ABCB1 TT* genotype independently predicted MACEs in ACS patients treated with clopidogrel after PCI (OR = 6.58; 95% CI: 1.24–34.92; *p* = 0.03). In contrast, Park et al. [24] found that although the coexistence of *CYP2C19* poor metabolizer status (**2*/**2*, **2*/**3* or **3*/**3*) and *ABCB1 3435 TT* was a strong independent predictor of MACEs in Asian patients undergoing PCI (HR = 4.51, 95% CI: 1.92–10.58, *p* < 0.001), it did not provide additional prognostic value when combined with traditional risk factors. The small sample sizes and inconsistent replication efforts have made it difficult to draw firm conclusions. However, since the CYP2C19 enzyme and the ABCB1 transporter interact in the pharmacokinetics of clopidogrel, combined pre-treatment genotyping of both polymorphisms may serve as an important predictor of therapeutic failure and adverse clinical outcomes after PCI.

In addition to the genetic variants defining the “clopidogrel resistance genotype”, our study identified rs7439366 (c.802T>C), a missense variant in the *UGT2B7* gene, as being associated with a lower incidence of adverse events in clopidogrel-treated patients. This suggests that variability in UGT2B7 activity may influence the pharmacokinetics of clopidogrel and consequently its clinical efficacy. UGT2B7, a highly active UDP-glucuronosyltransferase (UGT) expressed predominantly in the liver and kidneys, plays a central role in the glucuronidation and detoxification of drugs, including clopidogrel (Appendix A) [25,26]. Random forest analysis also showed that the *CC* genotype for rs7439366, which is associated with increased expression of *UGT2B7* in some tissues (Appendix A), conferred protection against adverse events after clopidogrel treatment in patients with the *GG* genotype for rs4244285 (*CYP2C19*). Until recently, CYP and UGT enzymes were thought to function independently due to their different functions and opposite membrane topologies (Appendix A). However, since some compounds are sequentially metabolized (e.g., clopidogrel or morphine), it is reasonable to consider that both enzymes may interact and regulate each other’s function, contributing to inter-individual variability in plasma drug concentrations. Supporting this, Kahma et al. [26] demonstrated that the acyl-β-D-glucuronide metabolite of clopidogrel acts as a “mechanism-based inhibitor” of CYP2C8 in human liver microsomes. This suggests that the proximity and interaction between CYP and UGT enzymes may facilitate the interaction of glucuronide metabolites with CYP enzymes. In addition, co-expression of *UGT2B7* has been shown to suppress CYP3A4 activity, suggesting mutual regulation between these enzymes [27]. Further studies are needed to elucidate the impact of rs7439366 (*UGT2B7*) on intracellular clopidogrel metabolite concentrations and clinical outcomes, and to determine whether it should be included in the prescribing algorithm to improve prevention of adverse events after PCI.

To date, PGx studies have focused primarily on the pharmacokinetic and pharmacodynamic mechanisms underlying variability in drug response. However, incomplete knowledge of cellular pathways and gene interactions has left a significant portion of heritability unexplained, leaving many patients at high risk for recurrent adverse events. Our findings suggest that genetic variants involved in lipid metabolism, vascular inflammation, and endothelial dysfunction—key processes in atherosclerosis—may significantly influence predisposition to such events. Several studies have explored the genetic architecture of clopidogrel response beyond *CYP2C19*, identifying new variants associated with MACEs that are indeed found in genes involved in some of these processes. Liu et al. [28] discovered eight novel MACE-associated variants in Chinese ACS-PCI patients treated with DAPT, including rs17064642 (*MYOM2*) and rs140410716 (*ECHS1*), two key genes in cardiac function. By adjusting their results for *CYP2C19***2*, Verma et al. [5] identified rs15121616272 (*LINGO2*), previously associated with BMI and diabetes mellitus, and rs35464072 (*NR3C2*), previously associated with oxidized LDL-induced inflammation, associated with high on-treatment platelet reactivity in European patients. Yang et al. [29] found that rs7792678 in *THSD7A* was associated with clopidogrel resistance in African Americans by causing endothelial dysfunction responsible for coronary artery disease. In light of these findings and the available evidence in diverse populations, a paradigm shift in the classical PGx approach is warranted. This change is particularly relevant in CV diseases, where the complexity of MACEs and hemorrhagic events following PCI requires a broader approach that integrates the individual’s molecular and clinical profile for improved risk stratification. Future PGx research should focus not only on pharmacological response genes but also on those related to disease predisposition and underlying pathological processes, paving the way for personalized therapeutic strategies that effectively address the various mechanisms involved in the recurrence of CV events.

Despite the promising results, there are some limitations to this study that should be considered. First, sample size, a common challenge in PGx research due to difficulties in recruiting sufficient participants to achieve the necessary statistical power for detecting significant associations. Second, the size of the panel, and consequently the number of variants identified. However, one advantage of our panel is that it enables us to analyze non-coding regions that would have been overlooked in an exome analysis. Third, since this was an observational study focused solely on analyzing genetic variants, complementary approaches—such as measuring lipid levels—to assess the functional effects of the identified variants were not included. Fourth, since most of the participants were of European descent, caution should be exercised when interpreting the results and applying them to other populations. It is important to note that the frequencies of the variants in our study may differ from those in other Caucasian populations due to North-West African influences. Finally, the ability to identify genome-wide associations with individual clinical outcomes was limited due to the small number of observed events. Consequently, these events were aggregated into a composite variable. Therefore, our results should be considered highly exploratory and hypothesis-generating, underscoring the need for larger studies with greater statistical power before these findings can be incorporated into therapeutic decision-making processes.

## 4. Methods

### 4.1. Study Population

This observational, retrospective study comprised ACS-PCI-stent patients previously enrolled in a Spanish non-randomized interventional study [9]—including an “intervention arm” (PGx-guided antiplatelet therapy) and a “non-intervention arm” (standard therapy without PGx testing)—as well as a group of patients without structural CV disease recruited de novo for the present study.

Based on previous findings from our group [9], we decided to divide the new study population into three well-characterized groups for the investigation of secondary CV effects of antiplatelet therapy (a visual overview of this selection is provided in Appendix A):○G1 group: a cohort of ACS-PCI-stent patients on antiplatelet therapy (guided or not by genetic testing) who experienced MACEs and/or bleeding during one-year follow-up (*n* = 109). This group was divided into two subgroups:“intervention group” (G1.1), in which carriers of *CYP2C19* LoF alleles (**2* or **3*) and/or the *TT* risk genotype for *ABCB1 C3435T* received prasugrel or ticagrelor, while patients with normal *CYP2C19* and *ABCB1* gene function received clopidogrel (*n* = 50).“non-intervention group” (G1.2), in which all patients were primarily treated with clopidogrel, regardless of their genetic profile (*n* = 59).○G2 group: a cohort of ACS-PCI-stent patients on PGx-guided antiplatelet therapy who did not experience MACEs and/or bleeding (*n* = 135).○G3 group: a cohort of patients without structural CV disease (*n* = 99).

The efficacy endpoint was the composite of MACEs during the 12 months after PCI, defined as follows: CV death, ACS, stroke, rate of definite stent thrombosis, and/or need for urgent revascularization not related to stent thrombosis. Safety endpoints included major or minor TIMI (Thrombolysis in Myocardial Infarction) bleeding unrelated to coronary artery bypass grafting. Details of the inclusion and exclusion criteria for the study groups, as well as the variables collected, are summarized in Antúnez-Rodríguez et al. [2], and in more detail in Sánchez-Ramos et al. [9] and Dávila-Fajardo et al. [30].

### 4.2. Genetic Analysis

Genomic DNA was extracted from saliva samples (buccal swabs) according to the method of Freeman et al. [31], with modifications by Gómez-Martín et al. [32]. All participants provided written informed consent prior to sample collection, which was approved by the Biomedical Research Ethics Committee of the Province of Granada (PEIBA code 1874-N-18). All procedures were performed in accordance with the tenets of the Declaration of Helsinki.

A total of 343 libraries, enriched for the specific regions of interest (gene panel), were prepared using the KAPA HyperPlus and NimbleGen SeqCap EZ Library Prep kits (both from Roche, Basel, Switzerland) and sequenced paired-end at 75 pb sequence length in pools of 40 samples. For a more detailed description of the entire process of custom gene panel design, library preparation, and sequencing, as well as bioinformatics data processing and quality control, see Antúnez-Rodríguez et al. [2].

### 4.3. Statistical Analysis

A descriptive and frequency analysis was performed on the main demographic and clinical characteristics of the entire cohort of patients included in the study, as well as on the characteristics related to the treatment prescribed after hospitalization in the cohort of ACS-PCI patients. The various characteristics were compared between the groups defined in this study using the “tableone” package (version 0.13.2) in R (version 4.2.2), with *p*-values less than 0.05 considered statistically significant.

#### 4.3.1. Association Study

Once the genetic variants were identified and annotated, and the filters previously described by our group were applied (omitting the low minor allele frequency (MAF) and LD filters to retain a greater number of variants) [2], imputation was performed using the Michigan Imputation Server (Haplotype Reference Consortium reference panel). Next, we used the “GENESIS v2.30.0” package (in R) to perform the relationship, ancestry, and differential analyses.

The association analysis of the identified genetic variants between the different groups was performed by establishing different comparisons:○“intervention” vs. “non-intervention” (G1.1 vs. G1.2);○“event” vs. “non-event” (G1 vs. G2);○“case” vs. “control” (G1&G2 vs. G3).

In addition, we decided to stratify the phenotype related to the occurrence of events (“event vs. non-event”) based on the prescribed treatment (clopidogrel or prasugrel). Efficacy and safety endpoints were then analyzed separately when significant associations were found for the development of secondary CV events in any of the above comparisons.

Phenotype association tests were performed using post-imputation genotypic probabilities under an additive genetic effect model, using the frequentist likelihood score method. The covariates age and gender, along with the first two principal components of ancestry, were entered directly into the model for all comparisons. For the “case vs. control” comparison, smoking was also included as a covariate. Given the nature of this study and the number of variants analyzed, no formal multiple testing correction (e.g., Bonferroni or False Discovery Rate (FDR)) was applied, as this would have resulted in overly conservative thresholds that would limit biological interpretation. Instead, we set four levels of significance: genome-wide significance level, −log_10_(5 × 10^−8^); 95% confidence, −log_10_(0.05/N°rs); 90% confidence, −log_10_(0.1/N°rs); and “suggestive”, −log_10_(1/N°rs), where “N°rs” denotes the total number of analyzed polymorphisms.

#### 4.3.2. Random Forest Analysis

Random forest is a supervised machine learning algorithm that combines multiple decision trees to generate more robust and accurate predictions. This method is used to classify input data into predefined categories based on a set of training data whose category membership is known. It also provides an unbiased selection of the variables that contribute most to the classification of the proposed model [33].

We performed a random forest analysis using the most relevant clinical variables in combination with the results of the association analysis (see Appendix A, p. 6) to assess their ability to predict susceptibility to the development of secondary CV events. This analysis was performed only for the “event vs. non-event” comparison, as it was considered the most relevant for applying the results to improve the decision algorithm used in clinical practice. Therefore, three models were generated:○Event vs. non-event” comparison, regardless of the antiplatelet drug received;○“Event vs. non-event” comparison in patients taking clopidogrel;○“Event vs. non-event” comparison in patients taking prasugrel.

For each model, each dataset was randomly divided into two groups in a controlled manner to ensure that all study groups were adequately represented. One was used to train the model (“training group”, ~70% of the samples), while the other was used to test the model and assess its predictive ability (“testing group”, ~30% of the samples). Discriminative performance indicators used to assess the classification efficiency of each model included ROC curves and their associated statistics: AUC and Youden index. In addition, other classification performance indicators such as sensitivity, specificity, and balanced accuracy were used.

## 5. Conclusions

This study is the first to design and analyze a gene panel that combines clinically relevant PGx loci with loci associated with major ischemic and/or hemorrhagic events related to antiplatelet therapy. We identified novel variants in clopidogrel-treated patients, including *ABCA1* rs2472434 and *UGT2B7* rs7439366. These variants, along with well-characterized ones such as *CYP2C19* LoF alleles and *ABCB1 C3435T*, may contribute to variability in drug response and susceptibility to secondary CV events. Our findings suggest that unexplained variability in clopidogrel response may stem from both suboptimal drug metabolism and susceptibility to the underlying disease, highlighting a paradigm shift in PGx research. Future studies should investigate not only genes that influence pharmacological responses, but also genes related to disease progression. However, given the limitations of this study, these results should be interpreted with caution, and further research is needed to clarify the influence of these candidate genes on the individual risk of secondary CV events after DAPT.

## Figures and Tables

**Figure 1 ijms-26-09766-f001:**
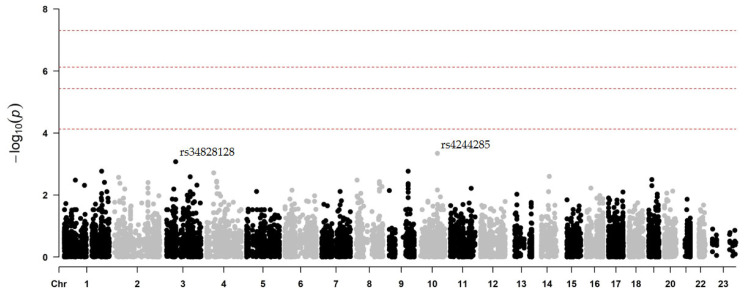
Manhattan plot of association results for the development of secondary CV events after clopidogrel treatment. In this comparison, 168 patients taking clopidogrel underwent phenotypic association testing. The chromosomal position is on the x-axis and the −log_10_ of the associated *p*-value is on the y-axis. The four significance levels considered are indicated in dotted lines: genome-wide significance level, −log_10_(5 × 10^−8^); 95% confidence, −log_10_(0.05/N°rs); 90% confidence, −log_10_(0.1/N°rs), and “suggestive”, −log_10_(1/N°rs). The most significant SNPs in this comparison are labelled. Taken from Antúnez-Rodríguez, 2025 [14].

**Figure 2 ijms-26-09766-f002:**
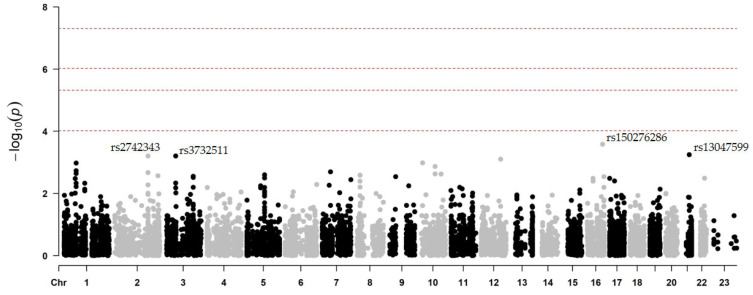
Manhattan plot of association results for the development of secondary CV events after prasugrel treatment. In this comparison, 73 patients taking prasugrel underwent phenotypic association testing. The chromosomal position is on the x-axis and the −log_10_ of the associated *p*-value is on the y-axis. The four significance levels considered are indicated in dotted lines: genome-wide significance level, −log_10_(5 × 10^−8^); 95% confidence, −log_10_(0.05/N°rs); 90% confidence, −log_10_(0.1/N°rs), and “suggestive”, −log_10_(1/N°rs). The most significant SNPs in this comparison are labelled. Taken from Antúnez-Rodríguez, 2025 [14].

**Figure 3 ijms-26-09766-f003:**
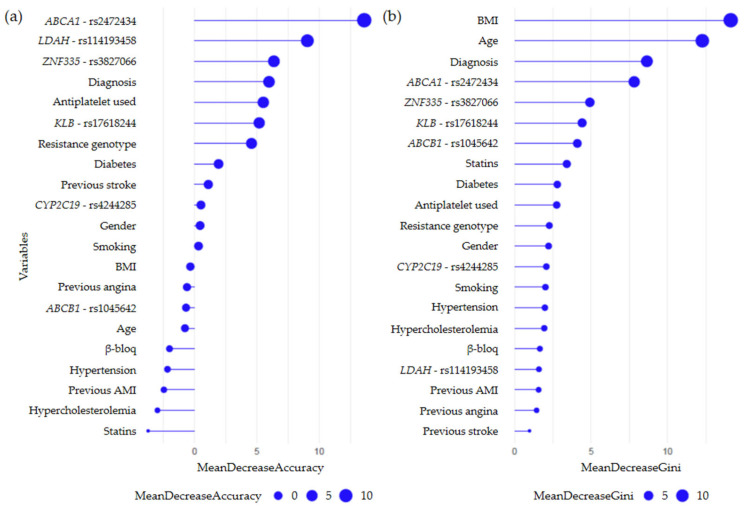
Variable importance plot from the random forest analysis, resulting from a model built around the development of secondary CV events during follow-up. In this analysis, random forest testing was performed on 243 patients. The variables are ordered top-to-bottom as most-to-least important in classifying between patients with an event and patients without an event. The ranked list of variables tells us the importance of each variable in classifying the data. (**a**) Mean Decrease Accuracy (MDA) plot. This graph is generated from each iteration (each decision tree generated) and shows which variables, if omitted, would have a greater impact on the loss of model accuracy. (**b**) Mean Decrease Gini (MDG) plot. This graph is interpreted in a similar way but is generated from the entire model rather than each iteration.

**Figure 4 ijms-26-09766-f004:**
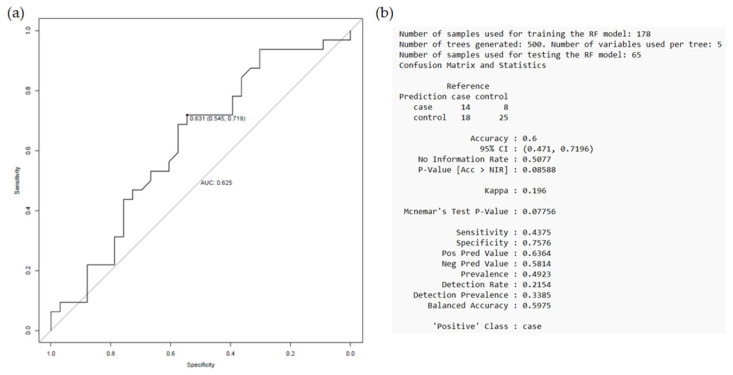
Model performance as established in the “event vs. non-event” comparison, regardless of prescribed treatment. The random forest model was trained on 178 samples and tested on 65 samples (total = 243 samples). (**a**) ROC curve with AUC and Youden index statistics, (**b**) parameters used to train the random forest model with the training set and the “confusion matrix” obtained from evaluating the model with the test set. Taken from Antúnez-Rodríguez, 2025 [14].

**Figure 5 ijms-26-09766-f005:**
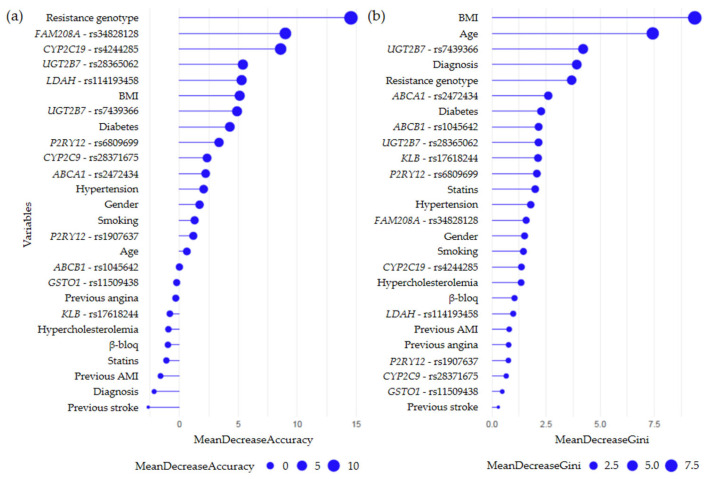
Variable importance plot from the random forest analysis, resulting from a model built around the development of secondary CV events following clopidogrel treatment. In this analysis, random forest testing was performed on 168 patients taking clopidogrel. Variables are ranked from most to least important in classifying patients with and without events. (**a**) MDA plot, generated from each iteration. (**b**) MDG plot, generated from the entire model rather than each iteration.

**Figure 6 ijms-26-09766-f006:**
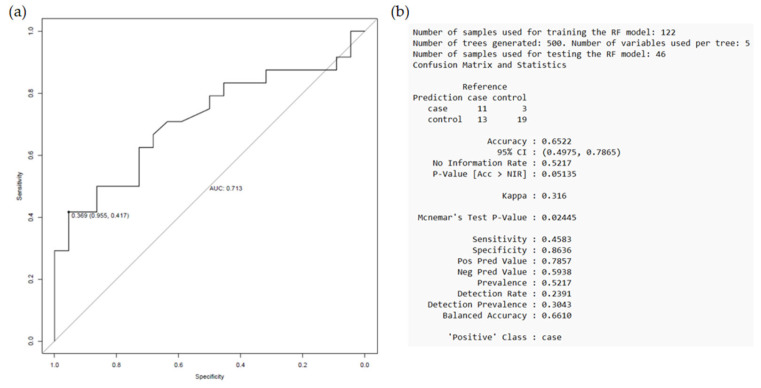
Model performance as established in the “event vs. non-event” comparison after clopidogrel treatment. The random forest model was trained on 122 samples and tested on 46 samples (total = 168 samples). (**a**) ROC curve with AUC and Youden index statistics, (**b**) parameters used to train the random forest model with the training set and the “confusion matrix” obtained from evaluating the model with the test set. Taken from Antúnez-Rodríguez, 2025 [14].

**Table 1 ijms-26-09766-t001:** Clinical and demographic characteristics of all study participants.

		ACS Patients			Controls	
	All (N = 343)	G1 (N = 109)	G2 (N = 135)	*p*-Value *	G3 (N = 99)	*p*-Value **
Age	65.9 (11.1)	66.8 (11.4)	64.8 (12.3)	0.202	66.4 (8.8)	0.339
Gender						
Male	232 (67.6)	78 (71.6)	101 (74.8)	0.670	53 (53.5)	**0.002**
Female	111 (32.4)	31(28.4)	34 (25.2)		46 (46.5)	
BMI	28.6 (4.6)	28.9 (4.6)	28.4 (4.5)	0.369	NA	NA
Ethnic origin						
Caucasian	335 (97.7)	105 (96.3)	132 (97.8)	0.517	98 (99.0)	0.423
Gypsy	6 (1.7)	3 (2.8)	3 (2.2)		0 (0.0)	
Moroccan	2 (0.6)	1 (0.9)	0 (0.0)		1 (1.0)	
Cv history	93 (27.1)	52 (47.7)	41 (30.4)	**0.008**	0	**<0.001**

Values are shown as *n* (%); for age and BMI, values are shown as mean (standard deviation). Baseline characteristics of patients were compared according to groups with the Chi-square test for categorical variables and Student’s *t*-test for continuous variables. Statistically significant results are shown in bold (*p* < 0.05). Abbreviations: ACS, acute coronary syndrome; BMI, body mass index; Cv, cardiovascular; G1, group 1 of ACS patients with secondary CV event; G2, group 2 of ACS patients without secondary CV event; G3, group 3 of controls without Cv disease; NA, not applicable. * This *p*-value compares G1 vs. G2. ** This *p*-value compares all ACS patients (G1 + G2) vs. controls (G3). Taken from Antúnez-Rodríguez et al., 2024 [2].

**Table 2 ijms-26-09766-t002:** Characteristics of treatment prescribed to ACS patients upon admission.

	All (N = 244)	G1 (N = 109)	G2 (N = 135)	*p*-Value
Acetylsalicylic acid	244 (100)	109 (100)	135 (100)	1.000
Antiplatelet therapy				
Clopidogrel	169 (69.3)	82 (75.2)	87 (64.4)	**0.035**
Prasugrel	73 (29.9)	25 (22.9)	48 (35.6)	
Ticagrelor	2 (0.8)	2 (1.8)	0 (0.0)	
DAPT duration				
1 month	14 (5.7)	11 (10.1)	3 (2.2)	**0.006**
3 months	2 (0.8)	2 (1.8)	0 (0.0)	
6 months	2 (0.8)	2 (1.8)	0 (0.0)	
12 months or more	226 (92.6)	94 (86.2)	132 (97.8)	

Values are shown as *n* (%). Prescribed treatment characteristics were compared according to groups with the Chi-square test. Statistically significant results are shown in bold (*p* < 0.05). Abbreviations: DAPT, dual antiplatelet therapy; G1, group 1 of ACS patients with secondary CV event; G2, group 2 of ACS patients without secondary CV event. Taken from Antúnez-Rodríguez et al., 2024 [2].

**Table 3 ijms-26-09766-t003:** Summary of identified variants, functional impacts, and associated biological processes.

Comparison	SNP	Alleles	Gene	Functional Consequence	Biological Process Involved
Intervention vs. non-intervention	rs72934556	T>G	*NBEAL1*	synonymous	Lipid metabolism
	rs2289843	A>T	*KALRN*	splice region	Endothelial dysfunction, atherosclerosis
Event vs. non-event	rs2472434	A>C	*ABCA1*	intronic	Lipid metabolism, inflammation
	rs17618244	G>A	*KLB*	missense	Lipid metabolism
	rs3827066	C>T	*ZNF335*	intronic	Lipid metabolism
Case vs. control	rs11076799	G>A	*ADCY9*	intronic	Endothelial dysfunction
	rs5370	G>T	*EDN1*	missense	Endothelial dysfunction, inflammation, lipid metabolism
	rs1800566	G>A	*NQO1*	missense	Inflammation, atherogenesis

**Table 4 ijms-26-09766-t004:** Annotation of the top-ranked SNPs identified in the association analysis for the development of events after clopidogrel treatment.

SNP	Chr	Position	Ref	Alt	Gene	Gene Role	Func.	MAF	*p*-Value	Beta	SD
rs4244285	10	96541616	G	A	*CYP2C19*	drug metabolism	synon.	0.15	0.00046	−1.9297	0.5785
rs4986894	10	96522365	T	C	*CYP2C19*	drug metabolism	upstream	0.15	0.00046	−1.9297	0.5785
rs34828128	3	56667213	T	C	*FAM208A*	unknown	synon.	0.02	0.00085	−2.2830	0.7476
rs2472434	9	107623249	A	C	*ABCA1*	disease predisposition	intronic	0.28	0.00171	−0.8835	0.2836
rs17618244	4	39448529	G	A	*KLB*	disease predisposition	missense	0.19	0.00194	0.8797	0.2859
rs4986938	14	64699816	C	T	*ESR2*	disease predisposition	UTR3	0.38	0.00251	−0.6907	0.2300

Genomic position is according to GRCh37/hg19 assembly. MAF was obtained from the Genome Aggregation (gnomAD)—Exomes Database report for Europeans. In this comparison, phenotype-association testing was performed on 168 patients taking clopidogrel under an additive genetic effect model using the frequentist likelihood score method implemented in the GENESIS v2.30.0 package. Abbreviations: SNP, single nucleotide polymorphism; Chr, chromosome; Ref, reference allele; Alt, alternative allele; Func, functional effect; MAF, minor allele frequency; Beta, beta coefficient corresponding to the effect size measure; SD, standard deviation; synon, synonymous variant; UTR3, 3′ untranslated region. Modified from Antúnez-Rodríguez, 2025 [14].

**Table 5 ijms-26-09766-t005:** Annotation of the top-ranked SNPs identified in the association analysis for the development of events after clopidogrel treatment, limited to genes involved in clopidogrel metabolism.

SNP	Chr	Position	Ref	Alt	Gene	Func.	MAF	*p*-Value	Beta	SD
rs2235048	7	87138511	G	A	*ABCB1*	intronic	0.47	0.03322	0.6291	0.2961
rs28365062	4	69964271	A	G	*UGT2B7*	synon.	0.14	0.00917	0.8886	0.3429
rs7439366	4	69964338	T	C	*UGT2B7*	missense	0.48	0.03090	0.5158	0.2394
rs4244285	10	96541616	G	A	*CYP2C19*	synon.	0.15	0.00046	−1.9297	0.5785
rs4986894	10	96522365	T	C		upstream				
rs11509438	10	106027059	G	A	*GSTO1*	missense	0.03	0.02745	−1.4006	0.6430
rs1907637	3	151104838	A	G	*P2RY12*	upstream	0.87	0.01814	0.9126	0.3876
rs6809699	3	151056598	A	C	*P2RY12*	synon.	0.83	0.03925	0.6609	0.3215
rs2046934	3	151057642	G	A		intronic				
rs10935838	3	151058247	A	G		intronic				

Genomic position is according to GRCh37/hg19 assembly. MAF was obtained from the Genome Aggregation (gnomAD)—Exomes Database report for Europeans. In this comparison, phenotype-association testing was performed on 168 patients taking clopidogrel under an additive genetic effect model using the frequentist likelihood score method implemented in the GENESIS v2.30.0 package. Only genetic variants with significant values (*p* < 0.05) are shown, grouped by gene and ordered by the direction of the pathway shown in Appendix A. Abbreviations: SNP, single nucleotide polymorphism; Chr, chromosome; Ref, reference allele; Alt, alternative allele; Func, functional effect; MAF, minor allele frequency; Beta, beta coefficient corresponding to the effect size measure; SD, standard deviation; synon, synonymous variant. Taken from Antúnez-Rodríguez, 2025 [14].

**Table 6 ijms-26-09766-t006:** Annotation of the top-ranked SNPs identified in the association analysis for the development of events after prasugrel treatment.

SNP	Chr	Position	Ref	Alt	Gene	Gene Role	Func.	MAF	*p*-Value	Beta	SD
rs150276286	16	81902879	C	G	*PLCG2*	disease predisposition	synon.	0.01	0.00026	−3.6496	1.0675
rs13047599	21	34926260	C	T	*SON*	unknown	missense	0.67	0.00057	−1.3065	0.3844
rs2742343	2	179569436	A	G	*TTN*	disease predisposition	synon.	0.03	0.00062	−3.1466	1.0184
rs3732511	3	56766435	C	G	*ARHGEF3*	disease predisposition	synon.	0.12	0.00062	−1.8055	0.5398
rs2888805	12	104380734	G	A	*TDG*	unknown	missense	0.10	0.00079	−2.5707	0.8042
rs6686	10	12209752	T	C	*NUDT5*	unknown	synon.	0.50	0.00104	−1.2772	0.3958
rs61781311	1	66087620	G	T	*LEPR*	disease predisposition	intronic	0.17	0.00105	−1.3725	0.4255
rs2472434	9	107623249	A	C	*ABCA1*	disease predisposition	intronic	0.28	0.00572	−1.2671	0.4631
rs3827066	20	44586023	C	T	*ZNF335*	disease predisposition	intronic	0.17	0.02880	1.0502	0.4836

Genomic position is according to GRCh37/hg19 assembly. MAF was obtained from the Genome Aggregation (gnomAD)—Exomes Database report for Europeans. In this comparison, phenotype-association testing was performed on 73 patients taking prasugrel under an additive genetic effect model using the frequentist likelihood score method implemented in the GENESIS v2.30.0 package. Abbreviations: SNP, single nucleotide polymorphism; Chr, chromosome; Ref, reference allele; Alt, alternative allele; Func, functional effect; MAF, minor allele frequency; Beta, beta coefficient corresponding to the effect size measure; SD, standard deviation; synon, synonymous variant. Modified from Antúnez-Rodríguez, 2025 [14].

## Data Availability

The data underlying this article will be shared on reasonable request to the corresponding author.

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
