# Peer review of "Custom Gene Panel Analysis Identifies Novel Polymorphisms Associated with Clopidogrel Response in Patients Undergoing Percutaneous Coronary Intervention with Stent"

_ijms, 2025, doi:10.3390/ijms26199766_

Round 1

Reviewer 1 Report

Comments and Suggestions for Authors

Dear Authors,

Please find attached my comments on the manuscript. 

Other comments:

The supplementary file is detailed and useful to support the data in the main manuscript. I would suggest that it be divided a bit better and the references to this file from the main manuscript should be more specific so that a reader understands well where to look for the information. You know these files by heart but a first time reader has a hard time identifying the info they are searching for. 

Also, have you calculated the power you have to detect significant SNPs? usually GWAS and GWAS-like studies have higher number of patients enrolled to ensure statistical power.

Best,

Reviewer 2 Report

Comments and Suggestions for Authors

What is new in the study, the most important data (CYP2C19/ABCB1) have previously been reported, and therefore using the same study population subdivided in the same study groups provides a lot of redundant information.

Introduction

The name of transporter encoded by ABCB1 should be stated – P-glycoprotein

Methods

The groups classification criteria should be expanded in short (some of the references are not  available as open access, no 9).

It is not clear and directly stated in methods whether patients were on double antiplatelet medication after the PCI procedure, i.e. with acetylsalicylic acid (ASA) combination (in the exclusion criteria you state “presenting contraindication for taking acetylsalicylic acid …”, which suggests its clinical administration).

Population characteristics as Caucasian seem not to be fully adequate due to due to historical admixture from North-West African populations. As for the CYP2C19 gene, it shows significant variability in the Spanish population, with CYP2C19*17 allele frequencies around 17.1% and *2 allele frequencies higher in some populations but lower in Spain compared to North-Western Europeans.

Why ultrarapid metabolizer genotype was not specifically analysed and discussed CYP2C19*17.

Results

In the data analysis there is a high author overinterpretation, and from statistical methodology imperfect. You somehow arbitrary select the SNPs for further analysis, and no correction for multiple comparisons was implemented. A lot of important findings is incorporated in supplementary materials, and those clearly show the importance/or lack of the findings. Redundant data from previously published papers appear. The new findings should be presented.

Table 3 significance level should be added

Table 4, 5, 6 what do you mean by most important as they are not significant

Why in the original result chapter you include a lot of data which has already been published

How you can rule out variability in ASA responses.

Discussion

The discussion should focus on really new findings not the ones reported previously.

Round 2

Reviewer 1 Report

Comments and Suggestions for Authors

Dear Authors,

Thank you for the detailed comments and for the effort put into resolving my comments. Best of luck with your future projects!

Reviewer 2 Report

Comments and Suggestions for Authors

I do accept authors responses.